# Role of miRNAs in Rheumatoid Arthritis Therapy

**DOI:** 10.3390/cells12131749

**Published:** 2023-06-30

**Authors:** Yiping Zhang, Meiwen Yang, Hongyan Xie, Fenfang Hong, Shulong Yang

**Affiliations:** 1Key Laboratory of Chronic Diseases, Fuzhou Medical University, Fuzhou 344000, China; yipingzhang53@126.com (Y.Z.); 18379441123@139.com (M.Y.); 2Queen Mary School, Nanchang University, Nanchang 330006, China; 3Department of Physiology, Fuzhou Medical College of Nanchang University, Fuzhou 344100, China; 4Technology Innovation Center of Chronic Disease Research in Fuzhou City, Fuzhou Science and Technology Bureau, Fuzhou 344100, China; 5Department of Foreign Language, Fuzhou Medical College of Nanchang University, Fuzhou 344100, China; xhy18811556540@126.com; 6Experimental Centre of Pathogen Biology, Nanchang University, Nanchang 330031, China

**Keywords:** rheumatoid arthritis, microRNA

## Abstract

Rheumatoid arthritis (RA) is a chronic systemic inflammatory disease characterized by autoimmunity, synovial inflammation and joint destruction. Pannus formation in the synovial cavity can cause irreversible damage to the joint and cartilage and eventually permanent disability. Current conventional treatments for RA have limitations regarding efficacy, safety and cost. microRNA (miRNA) is a type of non-coding RNA (ncRNA) that regulates gene expression at the post-transcriptional level. The dysregulation of miRNA has been observed in RA patients and implicated in the pathogenesis of RA. miRNAs have emerged as potential biomarkers or therapeutic agents. In this review, we explore the role of miRNAs in various aspects of RA pathophysiology, including immune cell imbalance, the proliferation and invasion of fibroblast-like synovial (FLS) cell, the dysregulation of inflammatory signaling and disturbance in angiogenesis. We delve into the regulatory effects of miRNAs on Treg/Th17 and M1/M2 polarization, the activation of the NF-κB/NLRP3 signaling pathway, neovascular formation, energy metabolism induced by FLS-cell-induced energy metabolism, apoptosis, osteogenesis and mobility. These findings shed light on the potential applications of miRNAs as diagnostic or therapeutic biomarkers for RA management. Furthermore, there are some strategies to regulate miRNA expression levels by utilizing miRNA mimics or exosomes and to hinder miRNA activity via competitive endogenous RNA (ceRNA) network-based antagonists. We conclude that miRNAs offer a promising avenue for RA therapy with unlimited potential.

## 1. Introduction

Rheumatoid arthritis (RA) is a chronic systemic autoimmune disease characterized by symmetrical inflammatory synovitis and pannus formation. The progressive destruction of bone and cartilage caused by these pathological processes can result in joint deformity, stiffness and disability. In RA, the immune pathology can also affect extra-articular organs, contributing to pulmonary and cardiac dysfunction [1].

The formation of pannus is a driving pathological process in RA damage [2]. The normal synovium consists of a layer of fibroblast-like synovial (FLS) cells and macrophages. While, in the RA articulate, the hyperplastic FLS cells and macrophages with other recruited immune cells lead to the thickening of the synovium. At the same time, the dysregulated angiogenesis process produces neovascular dysfunction with a defective vessel wall. This impairs cells’ ability to deliver oxygen and creates a hypoxic environment. Eventually, the thickened synovium, highly dysregulated neovascularization, and inflammation collectively form a pannus (Figure 1) [3,4]. The pannus is a hypermetabolic lesion that requires large amounts of nutrients and oxygen to maintain its aggressiveness and tissue-destroying ability [5], ultimately leading to permanent disability [4,6].

While the exact etiology of RA remains elusive. The prevailing hypothesis is genetic–environmental interactions, in which the human leukocyte antigen (HLA)–DRB1 gene plays a crucial role, and provoking environmental triggers such as smoking, infection, and sexual hormones are risk factors incorporated in RA onset [7,8]. Epigenetics, which involves transcriptional factors, DNA methylation, histone modification and non-coding RNA (ncRNA), serves as a link between environmental stimuli and genetic regulation [9]. In particular, aberrant alterations of DNA methylation and microRNAs (miRNAs) disrupt the immune system and contribute to autoimmune and inflammatory disorders [10].

miRNAs are a group of small non-coding RNAs (ncRNAs) that undergo several steps of biogenesis. It starts with long primary miRNA (pri-miRNA) transcripts that are transcribed from DNA. These transcripts are then processed by class II RNaseIII nucleases (Drosha) into stem–loop precursor nucleotides (pre-miRNA). The next step involves cleaving pre-miRNA by class III RNaseIII nucleases (Dicer) into a double-stranded miRNA. Finally. This double-stranded miRNA is further modified into a single-stranded miRNA of about 18–22 nucleotides in length by RNA-induced silencing complex (RISC) cleavage. By forming Watson–Crick pairings with the target message RNAs (mRNAs) on the 3′ untranslated region (UTR), miRNAs can repress gene expression at the post-transcriptional level [11,12].

Despite advances in the pathophysiology of RA, there is still no effective method for effectively controlling the disease process. The main drugs used to treat RA are nonsteroidal anti-inflammatory drugs (NSAIDs), glucocorticoids (GCs) and disease-modifying antirheumatic drugs (DMARDs) such as methotrexate (MTX). Current therapeutic interventions, such as pharmacological agents and surgical procedures, can only mitigate symptoms or decelerate the disease progression and activity. These are costly and have numerous adverse effects [13,14,15]. Hence, identifying new treatment candidates has become an urgent focal issue.

During the pathogenesis of RA, researchers have discovered changes in the expression levels of various miRNAs in different tissues. These miRNAs play a role in both innate and adaptive immunity and inflammation, which are known to affect the onset and progress of RA [16,17,18]. Therefore, it can be inferred that measuring the expression level of miRNAs can serve as a biomarker for diagnosing, prognosing or monitoring the effectiveness of therapeusis for RA. Given that, this review aims to explore the mechanisms of miRNAs in the pathogenesis of RA, mainly with a focus on immunity cells such as T helper (Th) cells, monocytes and macrophages, inflammation, FLS cells and angiogenesis. The goal is to shed light on the role of miRNAs in RA therapy and identify new avenues for RA treatment.

## 2. Role of miRNAs in RA Development

A small RNA sequencing study compared the expression of 262 miRNAs in the plasma of RA patients and healthy controls. The results showed that there were 175 miRNAs that had significant differences in expression between the two groups. Further analysis of the differentially expressed miRNA panel revealed their involvement in various cell types related to RA pathogenesis, such as macrophages, FLS cells, endothelial cells and osteoarthritis [19]. miRNAs have attracted more and more attractions as an inducer or inhibitor in the development of RA.

The following section will delineate the role of different miRNAs by dividing them into cellular or tissue structures involved in RA pathophysiological progression. This may seem repetitious, but we considered the possibility that a single miRNA may be expressed in multiple cells and/or affect multiple processes involved in RA progression. Therefore, Table A1 Appendix A presents the changes in miRNA expression and their role in different cells of RA, hoping to provide a more comprehensive view of the disease from a different perspective and to provide new insights for diagnosis and treatment.

### 2.1. miRNAs in Cells of the Immune System in RA

The innate and adaptive immune systems both participate in the autoimmune response of RA. The immune cell contributors include T cells, B cells, macrophages, monocytes, dendritic cells and neutrophils. miRNAs regulate their cellular processes and influence the activation of these immune cells [20].

#### 2.1.1. T lymphocytes

In RA, T cells are the primary immune cells involved, which are mainly divided into CD4+ T helper (Th) cells and CD8+ cytotoxic T (CTL) cells. While some miRNAs do have an effect on CTL activity and immune system regulation [21], it is the Th1, Th17 and regulatory T (Treg) cells that play the prominent role in the pathophysiology of RA [22].

Treg cells, expressing CD4, CD25 and Forkhead box P3 (Foxp3), are essential for regulating the immune system by inhibiting the activation and proliferation of autoreactive T cells. Cytokines such as interleukin (IL)-2, transforming growth factor β (TGF-β), transcription factor FOXP3, and signal transducers and activators of transcription (STAT)-5 are required for Treg cell proliferation, activation and differentiation [23,24,25,26]. Natural Treg cells (nTreg) and induced Treg (iTreg) cells are the two broad categories of Treg cells. The depletion of these cells leads to the altered expression of miRNAs, such as miR-551b, miR-448 and miR-124, and the dysregulation of immune-associated pathways involved in RA [27]. In a widely used RA animal model, called the collagen-induced arthritis (CIA) model, extracellular vesicles (EVs) derived from iTregs have been found to be effective in inhibiting inflammatory infiltration, effector T cell proliferation, synovial hyperplasia, pannus formation and bone erosion by inhibiting Notch1 expression via miR-449a, which helps to restore the balance between Treg and Th17 cells. Moreover, the expression of pro-inflammatory cytokines, such as IL-6, IL-17A, tumor necrosis factor (TNF)-α and interferon (IFN)-γ, was inhibited, whereas that of anti-inflammatory cytokines, such as IL-10, was upregulated [28].

As a subset of the Th cell, the Th17 cell also plays an important role in the regulation of autoimmune diseases. In the process of Th17 differentiation and the production of pro-inflammatory cytokines like IL-17 and IL-22, the transcription factors STAT-3 and RAR-related orphan receptor gamma t (RORγt) work together with cytokines such as TGF-β, IL-6, IL-23, IL-21 and IL-1β [29,30]. These cytokines promote macrophages and FLS cells to produce factors like IL-1, IL-6, TNF and granulocytes macrophage colony-stimulating factor (GM-CSF), which recruit neutrophils and monocytes to the joint cavity and further aggravate synovial inflammation and articular damage [30]. By targeting the expression of RORγt and IL-17 mRNA, let-7g-5p [31], miR-26b-5p [32] and miR-124 [33] can interfere with Th17 differentiation from naïve CD4+ T cells and Treg cells. miR-124 represses the IL-6 signaling pathway by inhibiting IL-6R expression and subsequently blocks the downstream phosphorylation of STAT3. Let-7g targets Fas, which promotes Th17 differentiation by inhibiting STAT1. Maresin 1 regulates the proportion of Treg and Th17 via the expression of miR-21, which modulates the activity of Foxp3, RORγt, STAT3 and STAT5 [34,35]. Another miRNA, miR-144-3p, can inhibit the expression of hypoxia-inducible factor 1α (HIF1α) and abolish its function on Th17 and Treg differentiation by attenuating the induction of RORγt and the degradation of FOXP3 protein, respectively [36]. Thus, the overexpression of these miRNAs can regulate Th17 differentiation and further control Th17-associated autoimmune diseases like RA.

Obviously, the above highlights a crucial point in the pathogenesis of RA, which is the imbalance between Treg and Th17 cells. Naïve T cells, activated under a specific cytokine environment, can express Foxp3 or RORγt transcription factors and differentiate into Treg and Th17 cells, respectively [37]. In parallel, there is a transformation between them in which Treg cells can be transformed to a Th17-like phenotype in the presence of IL-6 and TGF-β [32], and Th17 cells can also convert into Treg cells during the resolution of inflammation [38].

Since the increase in Th17 cells and the defect of Treg are crucial mechanisms in RA immunopathology, how to restore their balance or control their conversion is one of the priorities in the treatment. It is easy to deduce that miRNA targets the mRNAs of Foxp3 and RORγt as they are crucial transcription factors for Treg and Th17 differentiation, as shown in Figure 2. The ratio of Treg/Th17 correlates with disease activity score 28 (DAS28), which reflects the activity of RA [34]. Considering these factors, initiating the expression of these miRNAs to restore Treg/Th17 balance could be one of the candidates for RA treatment.

In recent RA research, Th1 cells appear to receive less attention than Th17 and Treg cells. However, their role in RA is essential. Th1 cells are effector T cells that differ from naïve CD4+ T cells under the induction of a master transcription factor T-bet. They produce cytokines such as IFN-γ, IL-2 and TNF, enabling them to defend infection and participate in chronic inflammatory diseases such as RA [39]. Initially, RA was regarded as a disease mediated by Th1 cells [40]. Most Th cells in the synovial fluid of RA patients express chemokine ligand receptor 13 (CXCLR3), which is thought to be a Th1 cell surface marker [41]. In patients suffering from chronic inflammatory diseases, repetitively stimulated Th1 cells accumulate in inflamed tissues along with the expression of Twist1 [42], a transcriptional repressor that regulates Th cell metabolic adaptation in chronic inflammation [43]. Twist1 and T-bet upregulate miR-148a expression in Th1 cells that are repeatedly activated. miR-148a represses Bim expression and restores T-cell-intrinsic apoptosis and antiapoptotic B-cell lymphoma 2 (Bcl-2) function, which is inhibited by Bim [44]. As one of the most critical cytokines secreted by Th1 cells, the expression of IFN-γ can be regulated by Treg cells by regulating miR-146a targeted Stat1, which is downstream of the IFN-γ receptor. This inhibition partially but not completely alleviates the autoimmunity caused by Th1 cells [45].

Interestingly, an imbalance between Th1 and Th2 cells exists in RA patients. Th2 cells, which secrete IL-4, are relatively rare in RA patients and play an anti-inflammatory function. Th1 and Th2 cells have antagonistic effects on their differentiation and cytokine production. For instance, IL-4 expressed by Th2 cells can inhibit the proliferation of Th1 cells and counteract the pro-inflammatory effects of IFN-γ [46]. However, even the existing Th2 cells tend to exhibit a Th1-like phenotype by expressing CXCR3 [47]. Therefore, in addition to restoring Treg/Th17 balance, favoring Th2 differentiation or function and altering the Th1/Th2 ratio could also be considerable methods to slow down RA progression. Unfortunately, miRNA studies on the Th1/Th2 balance in RA are scarce.

Another controversial issue is whether Th1 or Th17 cells plays a more prominent role in RA pathogenesis [48], which affects the choice of therapeutic targets. miR-10b expression is upregulated in RA models accompanied with the disturbed balance of Th cells. By targeting *GATA3* and *PTEN*, miR-10b alters the balance between pathogenic and regulatory T cells, e.g., Th17 versus Treg cells and Th1 versus Th2 cells, resulting in an imbalance in CD4+ T cell polarization in RA patients, followed by a high expression of cytokines, mainly IFN and IL-17A. Th cells overexpressing miR-10b further activate FLS cells and macrophages in an inflammatory manner [49]. Inhibiting miR-10b expression could reduce inflammatory and immunopathological responses.

#### 2.1.2. Monocytes/Macrophages

Circulating monocytes and tissue macrophages are elevated and play a role in the development of inflammation in RA patients [50]. Monocyte chemoattractant protein-1 (MCP-1) and chemokine (C-C motif) ligand 2 (CCL2) are chemokines produced by osteoblasts and FLS that can bind to the receptors on monocytes and enhance their migration upon activation [51,52]. Cysteine-rich 61 (Cyr61 or CCN1) can upregulate CCL2 expression by repressing miR-518a-5p via the mitogen-activated protein kinase (MAPK) signaling pathway, triggering articular swelling and monocyte infiltration [51].

Ren et al. [53] reported that activated monocytes with CD14+ expression exhibit enhanced anti-apoptosis capacity. They suggested that this might result from the inhibition of high-mobility group box-containing protein 1 (HBP1/HMGB1) transcription by miR-29b, which impairs Fas-directed apoptosis. Additionally, substantial miRNA changes were found in CD14+ and CD16+ monocytes, in which the reduction in miR-27a-3p in CD14+ monocytes was related to carotid intimate media thickness (CIMT), and the decrease in miR-30c-5p, miR-124a-3p, miR-128-3p and miR-328-3p was related to the formation of atheroma plaques. These changes might contribute to cardiovascular damage in RA patients [54]. The downregulation of the miR-146a/*Relb* axis could contribute to osteoclastogenesis, leading to bone damage [55]. CD14-positive monocytes can transform into either M1-type macrophages (pro-inflammatory macrophages) or osteoclasts, depending on whether they express CD16 antigen or not, leading to synovial inflammation and bone erosion [52].

RA monocytes have a propensity to differentiate into dendritic cells (DCs) early on, as revealed by an ex vivo study [56]. Compared to healthy individuals, autoimmune patients have higher levels of DCs in their affected tissues but lower levels in their blood [57]. The predominant subsets of DCs infiltrating RA-SF are monocyte-derived DCs (Mo-DCs) or inflammatory DCs (infDCs). They can induce the differentiation of various CD4+ T cells depending on the inflammatory milieu. Mo-DCs produce TGF-β, IL-1β, IL-6 and IL-23 and induce Th17 differentiation, leading to RA damage such as chronic inflammation, cartilage erosion and bone loss [57,58,59]. IFN-γ can trigger DCs to secrete proinflammatory and Th1 cytokines [60]. Simultaneously, CD4+ T cells can release GM-CSF and IL-4 to assist Mo-DCs formation from DCs [61]. Mo-DC cells can transdifferentiate into osteoclasts, exacerbating the bone loss of RA [58]. These create a vicious cycle driven by Mo-DCs that progressively worsens the tissue damage in RA.

DCs promote Th17 cell differentiation by activating latent TGF-β via integrin αv. By targeting *ITGAV*, the gene encoding integrin αv, miR-363 can inhibit Th17 differentiation and cytokines [62]. A corresponding decrease in AXL, a tyrosine kinase receptor that blocks DC activation, was observed in Mo-DCs with increased miR-34a expression. Moreover, miR-34a can enhance major histocompatibility complex (MHC) II presentation from Mo-DCs to stimulate T cells, especially Th17 and Th1 cells. miRN-34a knockout mice showed reduced Th17 and IL-17 expression and less joint damage [63]. Additionally, exposure to aryl hydrocarbon receptor (AhR) ligands can induce monocytes to differentiate into Mo-DCs and activate AhR to enhance Th17 differentiation [58]. A study confirmed similar results by examining the effect of smoking on the AhR pathway in RA patients [64]. miR-223 can attenuate AhR-pathway-mediated inflammation [65].

Similarly, macrophages also play an important role in RA progression, especially the pro-inflammation phenotype (M1). Classically activated macrophages (M1-type) are induced by lipopolysaccharide (LPS), IFN-γ and TNF. M1-type macrophages are abundant in RA joints, and their activation initiates the transcription of NF-κB and hypoxia-inducible factor 1α (HIF1α), leading to the production of pro-inflammatory cytokines such as IL-1β and TNF-α and the phagocytosis of pathogens, whereas the secretion of IL-4 and IL-13 leads to alternatively activated macrophage (M2-type) formation via the activation of a Janus kinase (Jak)-3/ STAT-6 axis. M2-type macrophages can produce immunosuppressive factors such as IL-10, IL-13 and TGF-β and modulate the immune response, but their abundance in RA is relatively low [50,66]. In brief, RA patients show an impaired differentiation from monocytes into M2-like macrophages, resulting in a relative abundance of M1-type macrophages. The imbalance between M1- and M2-type macrophages contributes to RA patients’ chronic inflammation and joint damage.

miRNAs play a vital role in regulating the function and polarization of macrophages in RA [66]. miR-494 can repress tenascin-C (TN-C) function, an activator of Toll-like receptor 4 (TLR4), and is highly expressed in macrophages to initiate inflammation. The overexpression of miR-494 in macrophages can inhibit NF-kB and the expression of inflammatory mediators, thereby preventing inflammation infiltration and joint damage [67]. Similarly, Wang Y et al. [68] demonstrated that miR-548a-3p inhibited the TLR4/NF-κB axis in macrophages when incubated with LPS (Figure 3A).

In contrast, some miRNAs contribute to inflammation and the imbalance of macrophage polarization. Membrane TNF (mTNF), which is expressed on monocytes, is correlated with macrophage differentiation and RA disease activity. A decrease in M2-type macrophage differentiation accompanies its elevated expression in monocytes of RA patients. Paoletti A et al. proposed that the paradoxical high expression of miR-155 in M2-type macrophages may be associated with this situation [69]. However, despite M2-type macrophages having the anti-inflammatory ability, its function can be reversed by miR-221-3p by inhibiting the JAK3/STAT3 pathway. Transfection with miR-221-3p/miR-155-5p can even change the M2-type macrophages from a TLR4-induced anti-inflammation secretion profile to an M1-specific IL-12 secretion (Figure 3B) [70].

#### 2.1.3. B Lymphocytes

B lymphocytes are another crucial component of the immune system along with T lymphocytes. Following activation, these cells produce autoantibodies such as rheumatoid factors (RFs) and anti-citrullinated protein antibodies (ACPAs) that are important for diagnosing RA and indicating disease activity. Moreover, they also express several miRNAs, such as miR-16 and miR-150, that are involved in inflammation [71]. A study on the effects of methotrexate (MTX) for RA revealed that several miRNAs were dysregulated in CD19+ B lymphocytes, among which miR-155-5p is informative for newly diagnosed RA patients. These miRNAs mainly regulate B cell activation, differentiation and related signaling pathways. In particular, four genes that were frequently targeted by these miRNAs are *HMGA2*, *PTEN*, *IGF1R* and *AGO1*, all of which have been implicated in autoimmune diseases [72].

#### 2.1.4. Neutrophils

Neutrophils are another type of immune cell that significantly affect inflammation response. It has been detected that numerous miRNAs are lowly expressed in neutrophils of RA patients, leading to the increase in a series of mRNAs related to migration, inflammation and cell survival. The depletion of miRNAs such as miR-126, miR-148a and miR-223 may be related to the decreased activity of miRNA processing genes, the effect of ACPAs and the influence of some cytokines like TNF-α and IL-6 [73]. It has been mentioned that miR-223 can suppress neutrophil extracellular trap (NET) formation in adult-onset Still’s disease (AOSD) [74]. This might also account for the enhanced formation of NETs in RA patients along with the accumulation of NET remnant myeloperoxidase (MPO) [75]. However, this hypothesis needs verification.

### 2.2. miRNAs in Fibroblast-Like Synoviocytes in RA

Synovial hyperplasia is a major pathological feature of RA. It results from the abnormal proliferation of fibroblast-like synovial cells (FLSs), which are also called rheumatoid arthritis synovial fibroblasts (RASFs) in some articles [76]. The inflamed and immune-cell-infiltrated synovia activates the FLS cells in an aberrant methylation pattern, contributing to cartilage degradation and pro-inflammatory factor production. Thus, hyperplastic FLS cells are considered as a clue cell type of joint destruction, and their appearance often implies a more aggressive phenotype of RA [77,78,79].

Several articles have mentioned the metabolism-related problems caused by FLS cells. The excessive proliferation of FLS cells and the dysregulated neovascularization can increase glucose consumption and acidify the joint cavity by producing a large amount of lactic acid via the Warburg effect. The high concentration of lactic acid affects the physiological activities of immune cells and osteoclasts, which worsens the disease [6,80]. A similar viewpoint was also mentioned by Zhang M. et al. [81]. The study discovered that miR-34a-5p can regulate glucose metabolism and apoptosis resistance by targeting lactate dehydrogenase A (LDHA). However, the expression of miR-34a-5p is suppressed by lncRNA glucose transporter 1 (TUG1) in RA-FLS. The hypoxic, high-pressure and acidic environment that develops during RA progression further activates FLS cells, leading to a more aggressive phenotype [82]. Moreover, hypoxia inhibits osteogenesis and stimulates adipogenesis in FLS cells, which affects both osteogenesis and energy metabolism in RA patients. The role of miRNAs in this process is unclear, but miRNAs might be used to prevent or reduce hypoxia and its adverse effects [83]. Therefore, restoring levels of beneficial miRNA and utilizing them to repress FLS proliferation could potentially alleviate RA damage.

Connective tissue growth factor (CTGF) is a protein highlighted in the early stage of RA. It is upregulated in FLS cells and causes FLS cell proliferation, angiogenic activities, articulate damage and pannus formation [84]. miRNA-146a-5p can repress CTGF secretion by inhibiting the NF-κB/IL-6/STAT3 signaling pathway [85]. Furthermore, miR-17-5p also targets the IL-6/JAK/STAT pathway to combat inflammation and bone erosion [86].

Overproliferated FLS cells exhibit a metastatic and aggressive phenotype similar to cancer cells [87,88]. This phenotype is associated with the overexpression of sex-determining region Y-box protein 5 (SOX5), a transcription factor that regulates FLS cell migration and invasion. Whereas miR-15a/16 can bind to *SOX5* 3′UTR and repress its expression and that of some pro-inflammatory cytokines [89], miR-449a can directly target HMGB1 and inhibit HMGB1-induced FLS invasion, migration and autophagy, as well as IL-6 expression [90]. miR-708-5p and miR-141-3p can target Wnt3a and FoxC1, respectively, to inhibit the β-catenin signaling pathway, demonstrating a suppression of FLS proliferation, invasion and migration [91,92].

Additionally, it is valuable to mention that FLS cells, which are a type of bone-marrow-derived mesenchymal stem cells (BM-MSCs), have the capacity for multidirectional differentiation, such as osteogenic differentiation [93]. The Wnt/β-catenin signaling pathway is a critical pathway that promotes not only cell proliferation, invasion and metastasis but also osteogenic differentiation. The overexpression of miR-218 can suppress secreted Dickkopf-1 (DKK1), which is the inhibitor of the Wnt/β-catenin signaling pathway, inducing osteogenesis in RA-FLS [94]. In contrast, a gene called the receptor activator of nuclear factor-kappa B ligand (RANKL) is essential for osteoclastogenesis. RA-FLS cells express RANKL and activate osteoclasts and macrophages to cause bone and cartilage damage [95]. Unlike other previously described miRNAs biased towards delaying the RA process, miR-515-5p promotes FLS cell proliferation and the cell cycle process and resists apoptosis in RANKL-treated FLS cells. Other studies have revealed that miR-515-5p inhibits Wnt-1-induced secreted protein (WISP1), a gene that regulates the Wnt signaling pathway and the proliferation and differentiation of osteoblasts via the TLR4 signaling pathway [96].

When it comes to apoptosis, p53 is a key gene that regulates cell death. Mutated p53 is elevated in RA patients with a longer half-life, which confers apoptosis resistance to FLS cells [95]. Wu H et al. proposed that miR-34a could activate the ataxia telangiectasia mutated protein (ATM)/ATM and Rad3-Related (ATR)/p53 signaling pathway by inhibiting the expression of cyclin I, which belongs to the cyclin family and regulates cell cycles via cyclin-dependent kinase (CDK) 5 [97,98]. A study that investigated the therapeutic mechanism of the drug (5R)-5-hydroxytriptolide (LLDT-8) for RA found that LLDT-8 inhibited the high expression of miR-4478 in RA-FLS. Subsequent studies revealed that LLDT-8 upregulated the expression of lncRNA WAKMAR2, which acted as a miR-4478 sponge to restore E2F1/p53 signaling, thus inhibiting FLS cell proliferation and invasion [99].

In conclusion, miRNAs are involved in almost all biological activities of RA-FLS cells, such as energy metabolism, apoptosis, osteogenesis, proliferation, migration and invasion. They regulate various genes and signaling pathways that affect these processes (Figure 4).

### 2.3. miRNAs in Inflammation Underlying RA

In addition to autoimmunity, inflammation is another important pathological mechanism in RA. We describe the inflammation component separately here to emphasize its vital role in the pathogenesis of RA. Virtually almost all inflammatory cytokines and effector cells involved in RA are also essential elements of autoimmune responses. Aberrant miRNA expression affects the immune responses and inflammatory processes in RA.

Among the inflammation progression, NF-κB, a transcription factor classically composed of p50 (NF-κB1), p65 (RelA) and REL, might be the most prominent and most studied one. Its activation can be divided into three pathways: (1) stimulation by inflammatory cytokines, especially TNF-α and IL-1β; (2) the initiation of ligand binding to TLR; (3) phosphorylation by the stress signal IκB kinase complex (IKK) [100]. It can be inferred that miRNAs regulate the inflammation involved in NF-kB mainly by interfering with these activation pathways.

As we discussed in the Section 2.1.2, two different miRNA-mediated pathways regulate anti-inflammation and anti-osteoclastogenesis: the miR-548a-3p/TLR4/NF-κB axis [68] and the miR-146a/*Relb* axis [55], respectively. Another miRNA that exhibits an anti-inflammatory function is miR-766-3p, which is found in the blood circulation of RA patients. miR-766-3p suppresses inflammatory gene expression and inhibits NF-κB signaling via targeting mineralocorticoid receptor (MCR) in TNF-α-stimulated FLS cells [101]. miR-23a targets IKKα to suppress its downstream IL-17 expression, resulting in inhibiting NF-κB [102].

One study investigated the effect of combination therapy in RA and found that miR-26b and miR-20a inhibit glycogen synthase kinase-3 beta (GSK-β) and nucleotide-binding domain (NOD)-like receptor protein 3 (NLRP3), respectively, and negatively regulate the GSK-3β/NF-κB/NLRP3 pathway [103]. The NF-κB-related pathway is involved in the pathological mechanisms of RA, in which NLRP3 acts as a key inflammasome. NLRP3 is mainly induced in the synovia of RA and contributes to RA development [104]. By combining with its apoptosis-associated speck-like protein containing a caspase recruitment domain (ASC), the NLRP3 complex activates pro-caspase-1, enabling the maturation of IL-1β and IL-18, which further promotes TH17 differentiation and synovial inflammation [29,104,105].

miR-223 derived from exosomes can target the 3′ UTR of *NLRP3* mRNA and delay the inflammation triggered by macrophages [106]. Furthermore, miR-20a can repress NLRP3 complex formation and inhibit IL-1β expression to restore Treg/Th17 cell balance [107]. In addition to the NF-κB/NLRP3 signaling pathway, a RelA/miR-30a/NLRP3 axis exists in macrophages. miR-30a is a vital feedforward hub that inhibits the RelA-induced NLRP3 inflammasome, but its expression is inhibited by upregulated RelA during TNF-α-induced inflammation initiation [108]. Thus, it can be speculated that mimics or the activators of these miRNAs can demonstrate a protective function on RA, repressing inflammation and delaying bone damage.

The role of long non-coding RNAs (lncRNAs) in RA was also investigated in some studies, which identified miRNAs as a part of the lncRNA-miRNA-mRNA or competitive endogenous RNA (ceRNA) network. In this network, lncRNAs inhibit miRNAs and affect their target mRNAs. Upregulated lnc-NEAT1 exerts inflammation and enhances disease activity by repressing miR-21 and miR-125a expression [109]. Chen J et al. reported that lnc-NEAT1 also suppressed miR-129 and miR-204, leading to the activation of the MAPK/extracellular signal-regulated kinase (ERK) signaling pathway, which worsened FLS synovitis [110]. LncRNA PVT1 acted as a miR-145-5p sponge to upregulate IL-1 and IL-6 expression and activate NF-κB [111].

However, some miRNAs have a negative role in promoting inflammation. miR-21 restores Treg/Th17 cell balance in T cells, whereas it silences *SNF5* in FLS cells, leading to the activation of the NF-κB and phosphoinositide 3-kinase (PI3K)/protein Kinase B (Akt) signaling pathway and the enhancement of cell proliferation [112]. The dysregulation of miR-128-3p represses its target histone deacetylase 4 (HDAC4) and activates the AKT/ mammalian target of rapamycin (mTOR) pathway to disturb various activities of FLS cells [113]. Moreover, the inhibition of miR-147 attenuates synovial inflammation and joint destruction, which may be related to the expression of cytokines and *CCL* and *DEPTOR* genes [114].

### 2.4. miRNAs in Angiogenesis Underlying RA Pathology

Angiogenic imbalance is an important process in the development of RA and combines with hyperplastic proliferated synovial cells to form a pannus. The angiogenesis process is regulated by various angiogenic stimulators such as angiopoietin (Ang) and some angiogenic inhibitors such as angioarrestin and IL-12. They modulate the balance of angiogenic signaling molecules such as vascular endothelial growth factor (VEGF), sphingosine-1-phosphate (S1P), integrin, epidermal growth factor (EGF) and matrix metalloproteinases (MMPs)1/9 on epithelial progenitor cells (EPC). EPC migration and tube formation result in neovascularization [115,116].

miRNAs also participate in regulating angiogenesis by affecting the expression of pro-angiogenic or anti-angiogenic factors. These miRNAs are also called angiomiRNAs (angiomiRs) [115]. The downregulation of miR-525-5p induced by apelin (APLN) promotes Ang1-enhanced EPC angiogenesis [117].

Among the numerous regulators, VEGF is the most highlighted one and interacts with many angiogenesis-related signaling pathways. One of these pathways is the VEGF-S1P kinase 1 (SphK1)-S1P/S1P receptor 1 (S1PR1) signaling axis. By reducing miR-16-5p levels, S1P weakens its control of VEGF and allows EPC to form tubes and promote angiogenesis [118]. Additionally, miR-146a-5p can inhibit the secretion of VEGF and MMPs induced by extracellular matrix metalloproteinase inducer (EMMPRIN) in fibroblasts and monocytes. Interestingly, scientists have declared that NF-κB and JAK/STAT activation mediates the upregulation of miR-146a-5p via TNF-α/IL-6 expression and that the dysregulation of this pathway could decrease miR-146a-5p [119]. By targeting VEGF and MMP14 in FLS, miR-150-5p represses EPC tube formation, angiogenesis and FLS migration and invasion [120]. While miR-485-3p upregulates VEGF expression indirectly via the protein inhibitor of activated STAT3 (PIAS3)/STAT3 axis, its expression is inhibited by its sponge, circEDIL3, to alleviate arthritis symptoms [121]. Hence, modulating angiogenesis by miRNAs can be a potential strategy to reduce pannus formation and mitigate RA.

## 3. Role of miRNAs in RA Diagnosis and Therapy

Since we have mentioned many miRNAs and their expressions and functions in the pathogenesis of RA, it is critical to utilize them in RA treatment. In fact, many studies have explored the practical applications of miRNAs in therapy.

### 3.1. miRNAs as Biomarkers

Detecting positive RFs and ACPAs in patients is an essential diagnostic criterion for RA. However, not all patients have detectable autoantibodies, so new diagnostic methods and criteria are crucial to give an accurate diagnosis and risk classification. As an essential regulator of gene expression, miRNA dysregulation has been observed in RA patients of serum [122], plasma [123], synovial fluid [124] and peripheral blood monocytes (PBMC) [53]. These miRNAs are involved in many aspects of RA pathophysiology, and their expression levels are associated positively or negatively with some conventional indicators of disease activity or status. It implies that they can be used as reliable biomarkers to monitor disease activity and predict treatment outcomes. miR-548a-3p [44], miR-21 [62] and miR-125a [62] have negative correlations with the levels of RFs, C-reactive protein (CRP), erythrocyte sedimentation rate (ESR) and DAS28 score in serum, indicating their potential as biomarkers for disease activity monitoring. Furthermore, the high expressions of miR-143-3p, miR-145-5p and miR-99b-5p in early RA patients suggest a more severe erosion progression [3].

The expression of some miRNAs can predict the response to drug therapy. Circulating miR-10a, which is downregulated in RA patients, can serve as a diagnostic biomarker. Its upregulation in response to methotrexate (MTX) treatment indicates a good prognosis [92]. Conversely, the downregulation of miRNAs such as miR-132-3p, miR-146a-5p and miR-515-5p predicts a favorable clinical response to the MTX [93]. Low levels of miR-15a/16 in serum predict a poor response to DMARD therapy [111]. A miRNA expression profiling study in adjuvant-induced arthritis (AIA) identified eight miRNAs that were associated with RA pathogenesis and their target genes. Among them, miR-22, miR-27a, miR-96, miR-142, miR-223 and miR-296 indicated celastrol’s therapeutic effect [125].

### 3.2. Drug-Regulated miRNA

The miRNA-based therapies discussed here can be classified into two types: antagonists that suppress the expression of RA-enhancing miRNAs and miRNA mimics that restore the expression and function of RA-suppressing miRNAs. For instance, MTX can upregulate miR-877-3p to inhibit GM-CSF and chemokine (C-C motif) ligand 3 (CCL3), thus weakening the proliferation and migration of FLS cells [126]. Resolvin D1 can upregulate miR-146a-5p to suppress CTGF expression [85]. Moxibustion, a therapeutic tool in traditional Chinese medicine, can revert Treg/Th17 cell imbalance with miR-144-3p [36]. In the section on FLS cells, we discussed osteogenesis and mentioned two crucial molecules, Wnt and RANKL. Berberine, a herbal compound, can inhibit Wnt/β-catenin-signaling-induced FLS proliferation and RANKL-mediated bone erosion, thus achieving articular protection and pannus inhibition [127,128].

Another approach is to modulate the upstream regulators of miRNAs. miRNAs are in a complex ceRNA network with many lncRNAs or circular RNAs (circRNAs) upstream, which repress their expression. The function of miRNAs induced or suppressed by RA can be altered by modulating the expression of these miRNA sponges. For instance, astragaloside can reduce the expression of lncRNA LOC100912373, suppressing the function of miR-17-5p. This can restore the ability of miR-17-5p to target pyruvate dehydrogenase kinase 1 (PDK1) and inhibit its expression, reducing the proliferation of FLS cells [129]. Similarly, paeoniflorin blocks FLS cell proliferation and mobility, curbs the cell cycle and suppresses inflammation by regulating the circ-FAM120A/miR-671-5p axis [130]. On the other hand, some miRNA antagonists utilize the ceRNA network to repress miRNA expression. In addition to the lncRNA WAKMAR2/miR-4478/E2F1/p53 axis that mediates the effect of LLDT-8, as mentioned earlier, there are also other examples of miRNA sponge regulation in RA. These include the triptolide-regulated hsa-circ-0003353/microRNA-31-5p/CDK1 axis [131] and the tocilizumab-regulated lncRNA MIR31HG/miR-214-PTEN-AKT axis [132]. These two pathways have been identified to upregulate the corresponding miRNA sponges and relieve the inhibition of miRNAs on their target genes. As a result of this, there is a reduction in cell proliferation and cell cycle arrest and a decreased expression of inflammatory factors.

Interestingly, another intriguing idea that some scientists have proposed is to use plant-derived miRNAs as a dietary intervention for diseases like RA [133]. This method is considered relatively safe and convenient. However, many challenges and questions still need to be addressed, such as the amount and type of miRNAs in plants, their degradation and elimination in the body, and the duration of treatment.

### 3.3. Exosome-Derived miRNA

As we have discussed in the Section 2.4, miR-150-5p and miR-485-3p are two miRNAs that regulate the expression of VEGF, a key factor in blood vessel formation [120,121]. This modulation is achieved by delivering exogenous miRNAs or their corresponding miRNA sponges via small vesicles called exosomes, which transfer genetic material between cells.

Exosomes are a type of extracellular vesicles (EVs) that are produced by almost all cells. Their double membrane is invaginated by the plasma membrane and can easily cross biological barriers to reach their target destination [134]. They can carry various cellular components, such as miRNAs, to mediate intercellular communication or to excrete metabolites [135]. Exosomes that carry miRNAs specific to RA have therapeutic potential for RA patients. EVs can mediate intercellular communication by transferring miRNAs between cells. The encapsulation of miRNAs in exosomes can protect them from degradation in the extracellular milieu.

A comprehensive miRNA screening of plasma exosomes from RA patients revealed 14 exosomal miRNAs that were abnormally expressed. Among them, miR-204-5p was significantly downregulated, and its level was inversely correlated with disease parameters such as RF, ESR and CPR. miR-204-5p can be transferred by T-cell-derived exosomes and can target FLS cells to inhibit their proliferation [136]. A novel cell-free replacement therapy based on exosomes derived from mesenchymal stem cells (MSCs) has attracted more interest. For example, miR-34a induced by bone marrow, (BM)-MCS-EVs, inhibits FLS cell proliferation and promotes apoptosis via the p53 pathway [98]. Additionally, BM-MCS-EVs deliver miR-223 to target NLRP3 and ameliorate inflammation [106]. The use of EVs generated from the autologous MSCs of the recipient can minimize the possibility of rejection.

Moreover, there are many research models or profiles for miRNAs to validate drug efficacy or to reach an early diagnosis, providing a possibility for precision therapy. We can foresee the potential of miRNAs in various aspects of RA treatment.

## 4. Discussion

miRNAs play a crucial role in different aspects of RA pathology and offer potential therapeutic and diagnostic applications for RA. Many existing drugs modulate miRNAs to target specific genes. Moreover, miRNAs can be secreted via exosomes and can be detected in almost all body fluids [135]. Detecting miRNA changes can provide an earlier diagnosis and a more precise and personalized treatment plan. Using miRNA to monitor the treatment regimens also facilitates the subsequent adjustment of regimens and the initiation of possible alternative treatment plans for patients who are refractory to conventional treatment regimens. Therefore, there is a need to identify more precise detection loci and utilize them as alternative diagnostic tools for disease progression monitoring, prognosis estimation and efficacy determination. With the wider use of liquid biopsies to detect miRNA, a relatively non-invasive means of early diagnosis and assessing the therapy efficacy is more available [137].

Despite the promising prospects of using miRNA in the diagnosis and treatment of RA, there are still many challenges and questions that still need to be addressed. Further research is required to clarify the causal connection between miRNAs and RA pathways and to identify potential targets for modulating the RA pathogenesis. It would be valuable to explore if there are any miRNA biomarkers that are exclusive to RA rather than being common to other autoimmune inflammatory diseases. Another limitation of many studies is that they involve patients with long-standing RA, and these patients are usually already receiving medications. Therefore, biomarkers for early RA diagnosis and disease progression without treatments are still lacking.

While identifying the hub in the network of miRNAs is vital to target critical pathways for treatment, the role that this hub plays in the normal regulation of life should be addressed. For instance, both miR-20a and miR-26b can target genes in the NF-κB signaling pathway [103]. However, NF-κB also has physiological functions in mediating the normal immune response [138]. The complete inhibition of the inflammation process and immune cells will compromise the normal immune response. Therefore, finding a balance or tipping the scales in favor of RA treatment is a critical issue that needs to be addressed. Moreover, miR-223 has been shown to have opposite effects on FLS cells depending on its target genes. It can target NLPR3 to reduce inflammation and induce apoptosis in FLS cells [139] or target forkhead Box O 1 (FOXO1) to enhance FLS proliferation and arthritis [125]. These inconsistent findings may be due to the lack of standardization in the study conditions and the baseline characteristics across different studies.

Another challenge is to achieve the tissue-specific delivery of miRNA therapeutics to minimize off-target effects and achieve dose optimization [115]. Using MSC-EVs as a carrier seems to be a promising approach. Previous studies have demonstrated that MSC transplantation is an effective therapy for RA as it can reduce joint inflammation and destruction and delay pannus formation. However, the clinical application of exosomes is still in the research phase. The practical application of MSCs is limited mainly by their tumorigenic properties and low proliferative capacity [140]. Moreover, extracting MSCs from the patient is an invasive and painful process, with the risk of infection [141]. Additionally, there is an issue of miRNA purification, in which the exosomes should carry a large cargo of purified miRNA in quantities that can reach the concentration required for the therapeutic window.

The miRNA panel can facilitate the diagnosis of RA and elucidate its molecular mechanisms in various cell types. It could also provide more insights and directions for future research by applying miRNAs in cancer to inhibit angiogenesis or FLS hyperplasia and induce apoptosis. Conducting a more comprehensive study of the miRNAs in RA patients can offer more information in diagnosis and treatment as our understanding advances.

## 5. Conclusions

RA is a chronic inflammatory and autoimmune disease that involves the infiltration of immune cells, the hyperproliferation of FLS cells, the disruption of neovascularization and the dysregulation of various pro-inflammatory cytokines. These factors contribute to cartilage and bone erosion, reducing quality of life and increasing mortality risk. We investigate the role of miRNAs in RA pathogenesis, particularly in Treg/Th17 and M1/M2 balance, Mo-DC-cell-induced Th17 cell differentiation, the NF-kB/NLRP3 signaling pathway and neovascular dysfunction, as well as in FLS-cell-mediated energy metabolism disorder, apoptosis resistance, osteogenesis and mobility. miRNAs have potential therapeutic and diagnostic applications as biomarkers for diagnosis or treatment. Novel therapeutic strategies such as miRNA mimics, miRNA antagonists and exosome delivery are employed to modulate miRNA expression and achieve better outcomes with fewer side effects. miRNAs represent a promising avenue for RA therapy with unlimited potential.

## Figures and Tables

**Figure 1 cells-12-01749-f001:**
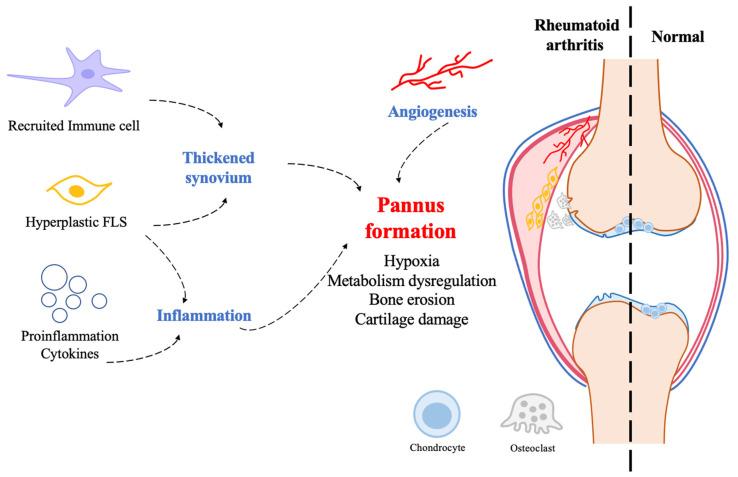
The formation of a pannus in RA. The thickened synovium filled with hyperplastic fibroblast-like synovial (FLS) cells and recruited immune cells, along with angiogenesis, participates in forming a pannus. The formation of a pannus implies an excessive proliferation of FLS cells along with dysfunctional neovascularization and the secretion of various inflammatory cytokines, resulting in an acidified, hypoxic and metabolically disturbed environment in the synovial cavity. The formation of pannus destroys bone and cartilage and ultimately results in permanent damage and disability.

**Figure 2 cells-12-01749-f002:**
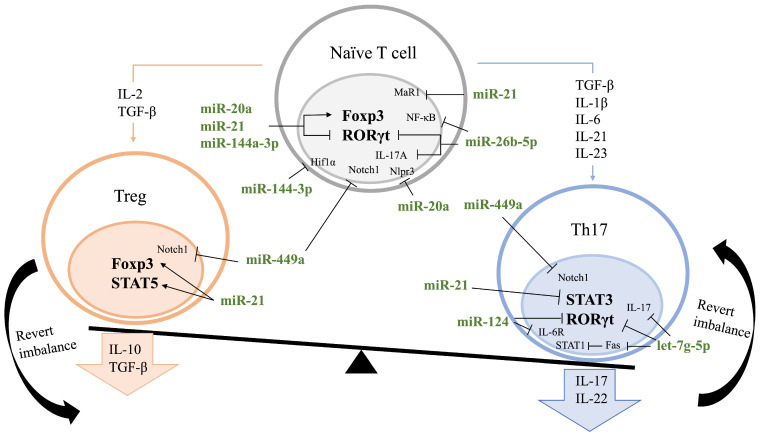
Mechanisms of miRNAs in restoring Treg/Th17 cell homeostasis in rheumatoid arthritis. miRNA, microRNA; TNF, tumor necrosis factor; TGF, transforming growth factor; IL, interleukin; Th, T helper cell; Treg, regulatory T cell; ROR-γt, RAR-related orphan receptor gamma t; NF-κB, nuclear factor κB; IL-6R, IL-6 receptor; STAT, signal transducers and activators of transcription; Foxp3, forkhead box P3; NLRP3, nucleotide-binding domain (NOD)-like receptor protein 3; Hif1α, hypoxia-inducible factor 1α.

**Figure 3 cells-12-01749-f003:**
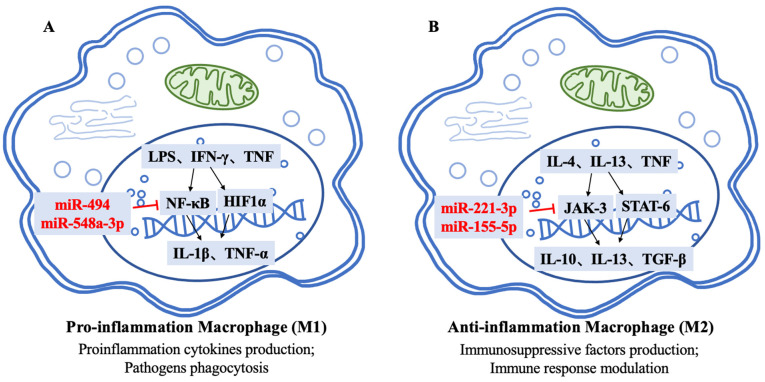
The role of miRNAs in the differentiation and function of macrophages. A schematic overview of miRNAs regulates the expression of key genes involved in Pro-inflammation Macrophage (**A**) and Anti-inflammation Macrophage (**B**) polarization. miRNA, microRNA; LPS, lipopolysaccharide; IFN, Interferon; TNF, Tumor necrosis factor; TGF, Transforming growth factor; IL, Interleukin; NF-κB, Nuclear factor κB; STAT, Signal transducers and activators of transcription; Hif1α, Hypoxia-inducible factor 1α; JAK, Janus kinase.

**Figure 4 cells-12-01749-f004:**
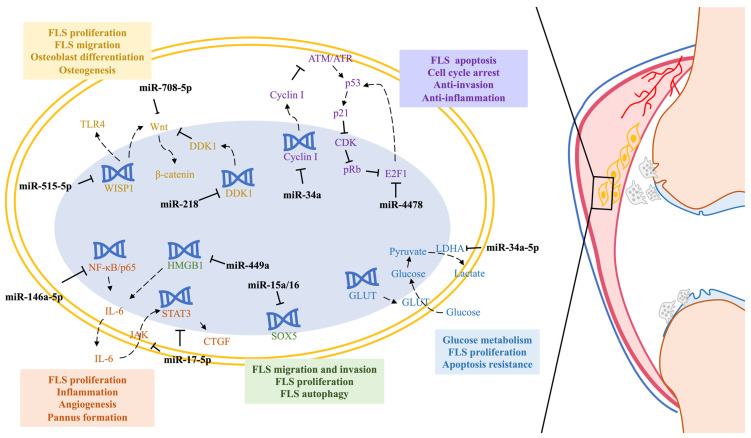
Schematic overview of miRNAs in RA-FLS cells. miRNAs are involved in targeting signaling pathways of metabolic disorders, metastasis, invasion, osteogenesis and apoptosis resistance in hyperplastic RA-FLS cells. RA, Rheumatoid arthritis FLS, fibroblast-like synovial; TLR4, toll-like receptor 4; WISP1, Wnt-1-induced secreted protein; CDK, cyclin-dependent kinases; IL, interleukin; NF-κB, nuclear factor κB; STAT, signal transducers and activators of transcription; JAK, Janus kinase; CTGF, connective tissue growth factor; HMGB1, high-mobility group box-containing protein 1; SOX5, sex determining region Y-box protein 5; LDHA, lactate dehydrogenase A; GLUT, glucose transporter; ATM, ataxia telangiectasia mutated protein; ATR, ATM and Rad3-Related.

## Data Availability

Not applicable.

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
