# Peer review of "Role of miRNAs in Rheumatoid Arthritis Therapy"

_cells, 2023, doi:10.3390/cells12131749_

Round 1
Reviewer 1 Report
In this manuscript, Zhang et al reported the functions of miRNAs in RA. The authors address both the miRNAs involved in the cells of the immune system, but also the miRNAs involved in the dysfunction of RA synoviocytes. The last section is dedicated to strategies for therapeutic targeting of miRNAs.
I have specific points to address:
- In the section entitled "3.1.2. Monocytes/Macrophages", authors should also address Mo-DCs. Indeed, this DC subset has been shown to be particularly abundant in the synovial fluid of RA patients and could be the main cellular source of Th17 cells induction. For review, please see: “Altered dendritic cell functions in autoimmune diseases: distinct and overlapping profiles” by Coutant F et al (PMID: 27652503) and “Shaping of Monocyte-Derived Dendritic Cell Development and Function by Environmental Factors in Rheumatoid Arthritis” (PMID: 34948462).
- The section entitled "3.1.1. T lymphocytes" discusses miRNAs in the context of Th17 and Treg, but nothing is said about Th1 cells, whereas, as it is written in line 150 "the main role in the pathophysiology of RA is played by Th1, Th17 and regulatory T (Treg) cells". The authors should address Th1 cells also in this section.
Typo :
Line 48 change « pri-miRNA » by pre-miRNA
Line 105: comma instead of period.
English language is correct, but the manuscript requires proofreading to rectify numerous typos.
Author Response
Reviewer 1:
Comments and Suggestions for Authors
In this manuscript, Zhang et al reported the functions of miRNAs in RA. The authors address both the miRNAs involved in the cells of the immune system, but also the miRNAs involved in the dysfunction of RA synoviocytes. The last section is dedicated to strategies for therapeutic targeting of miRNAs.
I have specific points to address:
- In the section entitled "3.1.2. Monocytes/Macrophages", authors should also address Mo-DCs. Indeed, this DC subset has been shown to be particularly abundant in the synovial fluid of RA patients and could be the main cellular source of Th17 cells induction. For review, please see: “Altered dendritic cell functions in autoimmune diseases: distinct and overlapping profiles” by Coutant F et al (PMID: 27652503) and “Shaping of Monocyte-Derived Dendritic Cell Development and Function by Environmental Factors in Rheumatoid Arthritis” (PMID: 34948462).
Answer: We gratefully thanks for the precious time the reviewer spent making constructive remarks. We have read the article you recommended and other related articles, and found that Mo-DCs can promote the differentiation of Th17 cells and can also be differentiated into osteoclasts. Therefore, we introduced DCs and explained its relationship with RA , as well as the regulation of miRNAs on the pathological process mediate by them.
- The section entitled "3.1.1. T lymphocytes" discusses miRNAs in the context of Th17 and Treg, but nothing is said about Th1 cells, whereas, as it is written in line 150 "the main role in the pathophysiology of RA is played by Th1, Th17 and regulatory T (Treg) cells". The authors should address Th1 cells also in this section.
Answer: We gratefully appreciate for your valuable comment. We have read the articles related to Th1 cells and realized that RA was initially considered to be a Th1-mediated disease. We understood the inflammation mediated by Th1 cells and the imbalance of Th1/Th2 ratio in RA. Based on this, we further added the role of miRNAs in Th1 cells in this text.
Typo :
Line 48 change « pri-miRNA » by pre-miRNA
Answer:Thank you for your careful consideration. We have reviewed the cited literature and related literature such as «The nuclear RNase III Drosha initiates microRNA processing », where pri-miRNA and pre-miRNA are also mentioned. We apologize for the misunderstanding caused by not introducing the full name, and we have further revised it.
Line 105: comma instead of period.
Answer: Thanks for your suggestion. We have revised the expression of the whole sentence, hoping to have a clearer and more formal expression.
Comments on the Quality of English Language
English language is correct, but the manuscript requires proofreading to rectify numerous typos.
Answer: Thank you for your suggestion. We have revised the description of the article and corrected some grammar and spelling errors. We hope this can make the article readable and more concise.
Reviewer 2 Report
Please see the attachment.

A large proportion of technical part of the review is described well. However, descriptions pertaining to general introduction and discussion are rather poorly written. At many places, words are used that are more suited for spoken English than for formal written English. Also, there are many grammatical and syntax errors in those sections of the manuscript. Therefore, the entire manuscript needs a thorough review to ensure proper usage of English language.
Author Response
Reviewer2
Comments and Suggestions for Authors
This is a review article that comprehensively described current literature on miRNAs as biomarkers, as mediators of cellular interactions involving different types of immune cells (e.g., T cell subsets, macrophage subsets, fibroblasts, etc.), and as potential druggable targets for rheumatoid arthritis (RA). Overall, this is an informative review for general readers as well as specialists in the field of rheumatology. Considering that it is a review article and that there are other reviews on miRNAs in RA, the topic is not original. However, this article has offered more insights into mechanistic aspects of miRNAs in different immune cells, as elaborated above. In that context, this article has addressed some mechanistic aspects not covered in other articles. However, there is a need for a thorough review of many parts of the manuscript for improvement of expression in the English language. Another concern is that there are omissions of some key publications that highlight a set of microRNAs (miRNAs) that are not covered in this manuscript. Their inclusion would make the article more thorough.
Michelle J Ormseth et al. Development and Validation of a MicroRNA Panel to Differentiate Between Patients with Rheumatoid Arthritis or Systemic Lupus Erythematosus and Controls. J Rheumatol 2020 Feb;47(2):188-196. doi: 10.3899/jrheum.181029.
Steven Dudics et al. The Micro-RNA Expression Profiles of Autoimmune Arthritis Reveal Novel Biomarkers of the Disease and Therapeutic Response. Int J Mol Sci 2018 Aug 5;19(8):2293. doi: 10.3390/ijms19082293.
Answer: Many thanks to you. According to you opinions we have revised the manuscript thoroughly and carefully, and also cited the articles you recommended to improve our manuscript. Thank you!
Comments on the Quality of English Language
A large proportion of technical part of the review is described well. However, descriptions pertaining to general introduction and discussion are rather poorly written. At many places, words are used that are more suited for spoken English than for formal written English. Also, there are many grammatical and syntax errors in those sections of the manuscript. Therefore, the entire manuscript needs a thorough review to ensure proper usage of English language.
Answer: We appreciate your review and suggestions. We have revised the introduction and conclusion sections of the article seriously, and adjusted the position of some paragraphs to make them more logical for reading. Regarding the language expression, we apologize for any inconvenience caused to you. Here, we re-examined the whole text and modified the grammar and expression of the article to make it more suitable for formal expression throughly. We hope these revision can make our manuscript better, thank you!
Round 2
Reviewer 1 Report
The authors have satisfactorily addressed most of my concerns.
Author Response
Response to Reviewer 1:
The authors have satisfactorily addressed most of my concerns.
Answer: We appreciate your positive feedback on our manuscript and your recognition of our work. We have also made some further revisions to address some minor issues and improve the quality of our manuscript. We hope that you will find our revisions satisfactory and acceptable.
Reviewer 2 Report
Comments are attached as suggestions for consideration of rewriting and/or deletion of some sentences. However, the entire manuscript requires another thorough revision and rewriting of many sections of the manuscript.
Important note: the Abstract/Summary in the online submission section should be replaced by the summary from the revised manuscript.

The current Abstract/Summary in the online submission section of the form should be replaced by that in the revised manuscript.
For the remaining manuscript:
There is much duplication of information in the first two sections of the manuscript, namely section 1, Introduction and section 2. Pathophysiology of RA. To make the text more focused, current section 1 can be deleted entirely and section 2 can be labeled as Introduction, followed by the section on miRNAs.
In addition, most of the text in the entire manuscript is still poorly worded. A large proportion of sentences are either incomplete or fragmented, with disordered phrases. That is another reason that authors should keep most of the miRNA information but shorten or delete unwanted general information especially in sections 1, 2, and 5.
Some suggestions for consideration for revision are given in the attached file in red font. But these cover only a fraction of the text, so a thorough revision of the entire manuscript by a writer proficient in English language is necessary.
Author Response
Response to reviewer 2:
Important note: the Abstract/Summary in the online submission section is very poorly written. It needs to be replaced by the summary in the revised manuscript.
Answer: Thank you for your careful consideration. We will submit the revised version of the abstract online.
For the remaining manuscript:
There is much duplication of information in the first two sections of the manuscript, namely section 1, Introduction and section 2. Pathophysiology of RA. To make the text more focused, current section 1 can be deleted entirely and section 2 can be labeled as Introduction, followed by the section on miRNAs.
Answer: We appreciate your thoughtful feedback. We have made revisions to the first and second sections of the manuscript, namely, "Introduction" and "Pathophysiology of RA". We have eliminated some redundant and irrelevant content and merged the two sections into one section "Introduction".
In addition, most of the text in the entire manuscript is still poorly worded. A large proportion of sentences are either incomplete or fragmented, with disordered phrases. That is another reason that authors should keep most of the miRNA information, but try to shorten or delete unwanted general information especially in sections 1, 2, and 5.
Answer: Thanks for your opinions! We have made revisions to sections 1, 2, and 5 carefully and seriously. We have merged two sections -- "Introduction" and "Pathophysiology of RA" into one section. We have removed the section "Conclusions and Perspectives" and replaced it with "Discussion." Besides, we have added "Conclusions" section according to the editor's suggestion. We hope that our revisions can provide a more concise discussion.
Some suggestions for revision are given below in red font. But these cover only a fraction of the text, so a thorough revision of the entire manuscript by a write proficient in English language is necessary.
Answer: Many thanks to you! We have the entire manuscript revised thoroughly according to your precious and detailed opinions. In addition, a thorough revision of the entire manuscript have been made by a write proficient in English language Thank you!
Line 18- is a kind of (replace kind with type)
Answer: Thanks for your suggestion! We have replaced the word ‘kind’ with ‘type’ in the manuscript.
Line 19-20 - The disturbance of miRNA (replace disturbance with dysregulation)
Answer: Thanks for your opinion! We replaced the word ‘disturbance’ with ‘dysregulation’ in the manuscript.
Line 22-23- expanding the way of RA diagnosis and treatment (this sentence may be deleted as it does not add new information)
Answer: Thanks for your suggestion! We have deleted the corresponding parts of the manuscript.
Line 32- RA treatment (change treatment to management)
Answer: Thanks for your suggestion! We have changed ‘treatment’ to ‘management’ in the manuscript.
Line 48, 49- Revise that sentence as follows: Furthermore, the immune pathology in RA can also involve extra-articular organs, resulting in pulmonary and cardiac dysfunction.
Answer: We have done what you suggested, thank you!
Line 54- Several potential factors have been implicated, such as cell-free DNA (cfDNA) [2] and macrophage polarization [3]. Delete this sentence- the two references cited here have no relevance to disease pathogenesis in RA.
Answer: Thanks for your suggestion! We have deleted the corresponding parts of the manuscript.
Line 66- achieve a radical solution
Revise this part of sentence to: achieve an effective control of the disease process.
Answer: Thanks for your suggestion! We have revised ‘achieve a radical solution’ to ‘achieve an effective control of the disease process’.
Line 67- producers; change it to procedures
Answer: It’s OK, ‘producers’ have been changed to ‘procedures’.
Line 85- DICER - Expand DICER and any other abbreviations on their first use.
Answer: Thanks for your suggestion! We expanded the abbreviations that first mentioned in our manuscript.
Line 106- Anebues- correct it to “avenues”
Answer: ‘Anebues’ have been changed to ‘avenues’.
Line 155- neonatal disorganized blood vessels lack an intact structural and tissue basis. Delete this sentence as the neonatal period has no relevance here.
Answer: Thank you for your advises! We are sorry for the potential confusion caused by our inaccurate wording. In fact , we stated that neovascular is involved in the formation of pannus in RA patients, not the neonatal period.
Line 189-190- Considering the belief that this division may fragment the expression of miRNAs in RA, however, a miRNA may be expressed simultaneously in multiple cells or processes of RA progression.
Replace this sentence with: This may seem repetitious, but we took into consideration the fact that a single miRNA may be expressed in multiple cells and/or affect multiple processes involved in RA progression.
Answer: We have done it as you suggested, thanks!
Line 198- revised title to: 3.1. miRNAs in cells of the immune system in RA
Answer: Thanks for your suggestion! We have changed the title to: 3.1. miRNAs in cells of the immune system in RA.
Lines 222-224- revise that sentence as follows:
and promotion of oncogenic pathways (delete highlighted part) immune-associated pathways involved in autoimmune diseases such as RA [40]
Answer: We have done it according to your suggestion, thanks!,
Lines 253-255- revise that sentence there to read as “Naïve T cells, activated under a specific cytokine environment can express Foxp3 or RORγt transcription factor and differentiate into Treg or Th17 cells, respectively [49].”
Answer: It has been revised according to your opinions, thank you!
Line 270-miRANs (correct it to miRNAs here and at few other places in the manuscript)
Answer: Thanks for your suggestion. We corrected miRANs to miRNAs through the manuscript.
Line 306-307- revise that sentence as follows:
By targeting GATA3 and PTEN, miR-10b alters the balance between pathogenic and regulatory T cells, e.g., Th17 versus Treg cells, and Th1 versus Th2 cells.
Answer: Thanks for your suggestion! We have revised the corresponding parts of the manuscript.
Line 373- M1-type macrophages express highly expressed in RA and , initiate---
Revise this sentence as: M1-type macrophages are abundant in RA joints and their activation initiates -------
Answer: We have revised it, thanks your!
Line 376- change Whereases to Whereas
Answer: We have changed “Whereases” to “Whereas”, thanks!
Line 390-391- The overexpression of miR-494 can inhibit the expression of inflammation factors and NF-κB induced by macrophages
Revise this sentence to: The overexpression of miR-494 in macrophages can inhibit NF-kB and the expression of inflammatory mediators
Answer: We have corrected it according to your opinions, thank you!
Line 417- They express autoantibodies such as RFs and ACPA
Revise it to: Following activation, these cells produce autoantibodies such as RFs and ACPA
Answer: Thanks for your suggestion! We have revised the corresponding parts in our manuscript.
Line 440- revise it to: However, this hypothesis remains specific verification
revise it to: However, this hypothesis needs verification
Answer: We revised it according to your suggestion, thank you!
Line 443: revise title to 3.2. miRNAs in inflammation underlying RA
Answer: We have revised it, thanks!
Line 447- inflammatory executive cells involved in RA:
Replace it with: effector cells involved in RA
Answer: The revision has been made, thanks!
Line 451- secrete numerous inflammatory factors and inflammasomes (delete highlighted part because inflammasomes are not secreted)
Answer: It has been done according to your suggestion, thanks!
Line 483- BMSCs (expand this and other abbreviations on first use)
Answer: It has been done according to your suggestion, thanks!
Line 524- revise title to: 3.3. miRNAs in angiogenesis underlying RA pathology
Answer: It has been done according to your suggestion, thanks!
Line 559- revise title to: 3.4. miRNAs in fibroblast-like synoviocytes in RA.
Answer: It has been done according to your suggestion, thanks!
This section should be moved with cells, right after immune cells (after line 442), which is just before discussion of inflammation (line 443).
Answer: We gratefully appreciate for your valuable comment. We have relocated the relevant section, namely, "miRNAs in fibroblast-like synoviocytes in RA" have been placed after "miRNAs in cells of the immune system". Thank you!
Figure shown in this section is not needed. It does not add any useful information over what is already shown in Figure 1
Answer: We gratefully thanks for your suggestions! However, we respectfully disagree with the suggestion to delete the figure, because we believe that it does not repeat the information presented in Figure 1. Figure 1 named “The formation of pannus in RA” illustrates the pathological process of pannus formation in RA patients, which involves excessive proliferation of FLS cells, recruitment of immune cells, inflammation and angiogenesis. The figure in question named “Schematic overview of miRNAs in RA-FLS cells” depicts the aspects of RA pathophysiology associated with FLS cells, and describes the relevant pathways that involve FLS metabolism, mobility, proliferation, apoptosis and osteogenesis. Thus present figure provides an important and complementary information to Figure 1 and should be retained in the manuscript.
Line 663- revise title to: 4. Role of miRNAs in RA diagnosis and therapy
Answer: Thanks! We have revised the title to: 4. Role of miRNAs in RA diagnosis and therapy.
Also, section 4.3 on biomarkers can be moved up before the section on miRNAs as drugs.
Answer: We gratefully appreciate for your valuable comment! " miRNAs as biomarkers " have been placed before " Drug-regulated miRNA ".
Line 668- or as miRNA analogs or antagonists. (delete this part of the sentence there, it has no connection there).
Answer: It has been deleted, thank you!
Line 718- 4.2. Exosome-derived miRAN (change to miRNA)
Answer: It has been corrected, thank you!
Line 789- 5. Conclusion and perspective
This section at present is lengthy and almost another summary of the whole article. It is not meant for that purpose. It can be shortened to 2 short paragraphs, maximum 3 paragraphs.
Only main conclusions to highlight some main aspects of miRNAs in RA should be mentioned here in one paragraph. Then perspective could be another paragraph.
Answer: Many thanks to you! The final section "Conclusions and perspectives," has been shortened according to your suggestions. As per the editor's request, the article requires a "Conclusions" section. Therefore, we have split the revised section into "Discussion" and "Conclusions".